# Vaccines against Drug Abuse—Are We There Yet?

**DOI:** 10.3390/vaccines10060860

**Published:** 2022-05-27

**Authors:** Benedict T. Bloom, Mary-Jessimine Bushell

**Affiliations:** Faculty of Health, University of Canberra, Canberra 2617, Australia

**Keywords:** vaccine, drug abuse, efficacy, ethical considerations

## Abstract

Background: Drug abuse is a worldwide problem that is detrimental to public health. The potential for drug abuse extends to both legal and illicit drugs. Drawbacks associated with current treatments include limited effectiveness, potential side effects and, in some instances, the absence of or concerns with approved therapy options. A significant amount of clinical research has been conducted investigating immunotherapy as a treatment option against drug abuse. Vaccines against drug abuse have been the main area of research, and are the focus of this review. Methods: An extensive search using “EBSCOhost (Multiple database collection)” with all 28 databases enabled (including “Academic Search Ultimate”, “CINAHL Plus with Full Text”, and MEDLINE), interrogation of the ClinicalTrials.gov website, and searches of individual clinical trial registration numbers, was performed in February and March of 2022. This search extended to references within the obtained articles. Results: A total of 23 registered clinical trials for treating drug abuse were identified: 15 for treatment of nicotine abuse (all vaccine-based trials), 6 against cocaine abuse (4 were vaccine-based trials and 2 were metabolic-enzyme-based trials), 1 against methamphetamine abuse (a monoclonal-antibody-based trial), and 1 multivalent opioid treatment (vaccine-based trial). As indicated on the ClinicalTrials.gov website (Home—ClinicalTrials.gov), the status of all but two of these trials was “Completed”. Phase 3 clinical trials were completed for vaccine treatments against nicotine and cocaine abuse only. Conclusion: Evidence in the form of efficacy data indicates that vaccines are not an option for treating nicotine or cocaine abuse. Efficacy data are yet to be obtained through completion of clinical trials for vaccines against opioid abuse. These findings align with the absence of regulatory approval for any of these treatments. This review further highlights the need for novel treatment strategies in instances where patients do not respond to current treatments, and while the search for efficacious vaccine-based treatments continues.

## 1. Introduction

Vaccines play a key role in maintaining and improving public health. They do so on a worldwide scale by providing protection against a range of bacterial and viral illnesses. The first vaccine was created over a century ago to provide protection against smallpox [1]. Since then, numerous vaccines have been created, including those most recently deployed in the fight against COVID-19.

Drug abuse, by contrast, is a worldwide problem that is detrimental to public health. The consequences of drug abuse include additional pressures on already strained public health systems, deterioration in the health and wellbeing of drug abusers and others—including unborn children—crime, and a myriad of social issues [2,3].

The potential for drug abuse extends to both legal and illicit drugs. These legal drugs include nicotine and alcohol, while illicit drugs include marijuana, the central nervous system (CNS) stimulants cocaine (including “crack” cocaine) and methamphetamine, and the opioid substances heroin and prescription pain medications [4].

Abused drugs are small, non-immunogenic molecules that remain undetected by the immune system, allowing them to cross the blood–brain barrier to the reward centers within the brain [3,5,6,7,8,9,10,11]. Current treatments for drug abuse include psychotherapy, pharmacotherapy, and combinations of both [1,12]. Pharmacotherapies include receptor agonists, partial agonists, antagonists, aversive therapy, and antidepressants. [3,9]. In the case of opioid-dependent patients, treatment options also extend to new, emerging drugs and psychological support [13].

Current psychotherapy and pharmacotherapy treatments have a number of potential drawbacks, including cost, limited effectiveness, patient non-compliance, social stigma and, in some instances, a disinclination by treatment providers regarding the use of some pharmacotherapies due to regulatory and abuse liability concerns [2,3,8,14,15,16,17]. Current treatments for smoking cessation have proved beneficial only in the short term [18]. For cocaine and methamphetamine, there is an absence of effective, approved pharmacotherapies [12,19,20,21,22,23,24]. For a number of pharmacotherapies, there are potential side effects.

The idea of an immunotherapy-based approach—including the use of vaccines—to treat addiction can be found in the literature as early as the 1960s and 1970s [1,25]. However, opiate receptor antagonists and methadone-based treatments have become the mainstay of treatment, as they are less demanding in terms of research, development, and investment requirements [25]. Over time, drawbacks associated with their use have begun to emerge, contributing to researchers once again looking into vaccines [25]. Recently, the school of thought that abused drugs are toxins has facilitated the exploration of additional pharmacotherapy and vaccine-based treatments [9,20,26].

Vaccines intended for treating drug abuse do so through active immunization by inducing the body to produce polyclonal antibodies in response to an antigen contained in the vaccine [1,3,10,12,17,22,26,27,28,29,30,31]. Vaccines for treating drug abuse typically consist of a hapten, a conjugate protein/macromolecular carrier, and an adjuvant [25,32]. A hapten is a structurally modified drug molecule to which a chemical linker has been attached [2,11,29,33]. The conjugate protein/macromolecular carrier is the component of the vaccine that is immunogenic, and is linked to the hapten. The overall structure is known as a hapten–conjugate protein/macromolecular carrier [1,2,3,6,9,11,14,15,16,17,21,22,23,25,26,29,32,33,34,35,36,37,38,39,40,41,42]. Adjuvants include aluminum-containing substances that enhance the immune response to an antigen, thereby producing stronger immunity [1,6,8,10,14,25,35]. Adjuvants are common in vaccines against drug abuse.

The aim of this literature review was to determine whether vaccines are an effective treatment option against drug abuse.

## 2. Materials and Methods

The PICOS criteria (Problem, Intervention, Comparison, Outcome, and Study Design) were applied to a literature search using “EBSCOhost (Multiple database collection)” with all databases (including “Academic Search Ultimate”, “CINAHL Plus with Full Text”, and MEDLINE) enabled. “Peer reviewed” and “English language” limiters were applied. Details of the search terms are shown in Appendix A. As shown in the table, 40 relevant articles were found.

A number of articles obtained in the search referenced “Clinical trial registration numbers”. Searches of these numbers within the ClinicalTrials.gov website [43,44,45,46,47,48,49,50,51,52,53,54,55,56,57,58,59,60,61,62,63,64,65] were undertaken to obtain details of the studies. In instances where study outcomes were not published on this website, additional searches of the individual clinical trial registration numbers (“NCT numbers”) and study titles were performed using EBSCOhost (with all databases enabled) and Google Scholar.

## 3. Results

Twenty-three registered clinical trials for treatments against nicotine, cocaine, methamphetamine, and opioid abuse were identified. All 15 of the nicotine trials, 2 of the cocaine trials, and 1 opioid trial involved vaccines. Two of the cocaine trials involved metabolic enzymes, and the methamphetamine trial involved monoclonal antibodies. An overview of the 18 trials of vaccines against drugs abuse is shown in Table 1. Table 2 contains a summary of the vaccines/treatments from these trials.

Defined inclusion criteria, exclusion criteria, and primary outcome measures were included for each clinical trial, with defined secondary outcome measures included for all but three of the trials (see Table 1).

Phase 1, 2, and 3 clinical trials were included in this review. “Phase 1” refers to “a phase of research to describe clinical trials that focus on the safety of a drug, usually conducted with healthy volunteers, and usually involving a small number of participants” [67]. “Phase 2” refers to “a phase of research to describe clinical trials that gather preliminary data on whether a drug works in people with a certain disease/condition. Safety continues to be a focus” [67]. “Phase 3” refers to “a phase of research to describe clinical trials that gather more information about a drug’s safety and effectiveness by studying different populations and different dosages and by using the drug in combination with other drugs. These studies typically involve more participants” [67].

## 4. Discussion

Vaccines against nicotine abuse: Phase 1, 2, and 3 clinical trials have been completed. A total of 15 trials were identified. Eleven trials were randomized controlled trials (RCTs), three were non-randomized controlled trials, and one was a cohort study. All RCTs and the cohort study were double-blinded. Six RCTs and the cohort study included multiple study centers to reduce bias. Inclusion and exclusion criteria, along with outcome measures, were clearly defined for each trial. Baseline characteristics were generally consistent across groups in each trial, further minimizing bias.

NicVAX: In the combined phase 1 and 2 trial NCT01318668, the vaccine was not efficacious in reducing brain activity from nicotine cues. Higher antibody levels were observed in the experimental group (*p* < 0.001) but, as indicated by measured exhaled carbon monoxide levels, did not significantly reduce cigarette use. The small sample size (*n* = 48), 31% dropout rate, and inclusion of only right-handed, 25- to 40-year-old males may have introduced bias (females were excluded because vaccine teratogenicity had not been assessed). Financial compensation provided to participants may have been a source of bias. No adverse effects were reported [68].

The phase 2 trial NCT00218413 (*n* = 51) assessed safety and antibody levels for varying vaccine doses. [27] states “increased abstinence related to Ab titer in phase II proof-of-concept”. Despite the study completion date of August 2006, the published study results were unable to be located [44].

In the phase 2 trial NCT00995033, combined treatment of a vaccine with standard treatment (varenicline and behavioral support) was not efficacious when compared with placebo and standard treatment. The attrition rate in both the experimental and control groups was approximately 25%. The large sample size (*n* = 558) would have reduced but not eliminated bias. Continuous smoking abstinence was not demonstrated for weeks 9 to 52 (“OR = 0.89, 95% CI = 0.62–1.29”), weeks 37 to 52 (“OR = 1.03, 95% CI = 0.72–1.46”), or weeks 9 to 24 (OR = 1.04, 95% CI = 0.74–1.47”) [69]. There was an insignificant difference between participants with the top 30% antibody levels and the control group for weeks 9 to 52 (“OR = 1.19, 95% CI = 0.71–2.00”). More adverse events relating to NicVAX were reported compared with the placebo (*n* = 45; 4.2% of events and *n* = 13; 1.2% of events, respectively). This was very highly statistically significant (*p* < 0.0001). Similar numbers of serious adverse events were reported in the NicVAX and placebo groups (*n* = 13; 1.2% of events and *n* = 15; 1.4% of events, respectively) [69].

In the phase 2 trial NCT00598325 (*n* = 74), when compared to previous studies, an additional dose of the vaccine was administered (i.e., six doses instead of five doses). The additional dose resulted in higher levels of antibodies, forming the basis for phase 3 trials [27,70].

In the phase 2 trial NCT00318383 (*n* = 313), the vaccine was administered using two different formulations and dosing schedules to determine which generated the highest antibody levels. This proof-of-concept-study showed that participants with high antibody response were more likely to abstain from smoking than participants who received placebo [27].

The phase 2 trial NCT00996034 assessed nicotine binding in the brain before and after vaccination, using tomography. In this single-arm study (*n* = 14), the dropout rate was approximately 20%. Nicotine binding was lower after vaccination (“mean nicotine binding = 54.9% and 49.1% before and after vaccination” [48], respectively), and was considered statistically significant (*p* < 0.05). No adverse events were reported [48].

The phase 3 trial NCT01304810 (*n* = 300) assessed antibody levels 24 months after the initial vaccine. Efficacy was not demonstrated [27]. Despite a study completion date of August 2011, final study data were unable to be located [49].

The phase 3 trials NCT01102114 (*n* = 1000) and NCT00836199 (*n* = 1000) assessed smoking cessation, abstinence, safety, and withdrawal symptoms. [50,51] The vaccine was well tolerated and safe, but not efficacious [27]. For NCT00836199, the abstinence rate in both the experimental and placebo groups was 11% [71]. Despite study completion dates of July and November 2011, respectively, final study data were unable to be located. [72,73] reported the vaccine was well-tolerated.

The phase 1 trial NCT01478893 (*n* = 82) was completed for SEL-068. Vaccine safety, tolerability, and immunogenicity were assessed in smokers and in non-smokers. While the trial was completed in March 2013, no results have been published [27].

The phase 1 trial NCT01672645 (*n* = 277) was completed for NIC7-001 and NIC7-003. The study assessed safety, antibody levels, and abstinence rates. Despite a study completion date of December 2015, the study’s results were unable to be located. This aligned with the literature, which indicated “not reported” and “ongoing” [27].

NIC002: The phase 2 trial NCT00736047 (*n* = 200) assessed vaccine efficacy by self-reported smoking status, exhaled CO levels, and immunogenicity. Primary endpoint was not meet in interim analysis [27].

The phase 2 trial NCT01280968 (*n* = 52) assessed changes in nicotine within the brain after “puffs of cigarettes” [55]. This study reported a 40% dropout rate. Results for vaccine-induced % change in nicotine area under the curve and % change in maximum nicotine concentration trended similarly in the experimental and placebo groups. There was an absence of probability values in the write-up; however, the findings did not support vaccine efficacy against the stated outcome measure. No serious adverse events were reported, though non-serious adverse events were reported (13 of 36 participants effected; 36% at risk, and 3 of 5 participants effected; 60% at risk, in the vaccine and placebo groups, respectively) [55].

The phase 2 trial NCT00369616 (*n* = 341) was completed for CYT002-NicQb. This study assessed smoking abstinence based on self-reporting and exhaled CO, immunogenicity, safety, and cravings via a questionnaire. Adverse events were reported in both the experimental and placebo groups. However, the odds ratio for a number of these events was equal to one, indicating equal risk. The vaccine was considered safe and immunogenic. At 2 months, there appeared to be a significant difference in abstinence between the experimental and placebo groups (47.2% and 35.1%, respectively; *p* = 0.036). However, efficacy was not observed between 2 and 6 months. After 12 months, a 20% difference in abstinence was observed between the participants with the highest antibody levels from the experimental group and the placebo group (*p* = 0.012), indicating a significant difference in efficacy. No questionnaire results were located. A total of 225 vaccine subjects (98.3%) and 104 placebo subjects (92.9%) experienced adverse events. This difference between groups was statistically significant (OR 4.3; 95% CI 1.3 to 14.7) [74]. While the early results showed promise, ultimately, the final findings were not conclusive.

The phase 2 trial NCT00633321 was completed for TA-NIC. This study assessed smoking quit rate after six months, antibody levels, and ongoing abstinence. The targeted number of 522 participants [57], and conducting the study across multiple sites, would have improved the internal validity of the dataset; however, the study has not yet been published. A summary statement in [27] indicates that efficacy was not demonstrated

Treatments against cocaine abuse: Phase 1, 2 and 3 clinical trials have been completed, and one phase 1 trial is recruiting. Six clinical trials were identified. All trials were RCTs, five of which were double-blinded. Three RCTs included multiple study centers, which reduced bias. Inclusion and exclusion criteria, along with outcome measures, were clearly defined for each trial. Baseline characteristics were generally consistent across groups in each trial, further minimizing bias.

TA-CD: In the phase 2 trial NCT00965263 (*n* = 15), patients received various doses of cocaine. Cocaine intoxication was self-reported, and heart rate and plasma cocaine levels were assessed. Five participants dropped out which, given the small sample size, would have affected bias. Variable antibody levels among participants were reported. In the low-antibody group, the effects of cocaine appeared to trend down from weeks 1 to 11, and plateaued at week 13. In the high-antibody group, the effects of cocaine trended down from weeks 1 to 13. Self-reporting by participants was another potential source of bias. Heart rate and plasma cocaine levels appeared to increase as the cocaine dose increased, indicating that the vaccine was not efficacious. For the low-antibody group receiving 0 mg of cocaine, plasma cocaine levels increased between weeks 3 and 13, which may be indicative of cocaine use outside the study. Overall, the vaccine did not demonstrate efficacy [58].

In phase 2 and phase 3 trials NCT00142857 (*n* = 115) and NCT00969878 (*n* = 300) respectively, [27,59,60] participants were randomized to receive either vaccine or placebo. Cocaine abstinence was assessed by screening urine samples. Peak antibody levels were measured. For NCT00142857 and as stated in [27], “high antibody response was associated with in-creased cocaine-free urine and reduced cocaine use compared with low antibody response and placebo groups”. For NCT 00969878, and as stated in [27] “there was no significant difference between placebo, low-antibody and high-antibody groups at week 16 in terms of cocaine-positive urine samples”. Overall, the vaccine was not efficacious. Serious adverse events were reported in both the vaccine and placebo groups 14 of 152 vaccine participants effected; 9.2%, and 15 of 148 placebo participants effected; 10.1%) [59,60].

The phase 2 trials NCT01846481 and NCT01887366 were completed for RBP-8000 and TV-1380, respectively [61,62]. These trials were enzyme-based, and were not the focus of this review.

The phase 1 trial NCT02455479 for aAD5GNE is recruiting. It will establish vaccine safety and measure cocaine, metabolites, and anti-cocaine antibodies in urine samples. The small sample size (*n* = 30) is consistent with a phase 1 trial [63].

Treatments against methamphetamine abuse: The phase 1 trial NCT01603147 for ch-mAb7f9 was completed [64,75]. This trial was monoclonal-antibody-based, and was not the focus of this review.

Vaccines against opioid abuse: The phase 1 trial NCT04458545 for Oxy[Gly]4-sKLH is recruiting (*n* = 45). The outcome measures will be vaccine safety and tolerability, and “drug liking” based on self-reporting [65].

## 5. Conclusions

Overall, this review found that, to date, despite the number of clinical trials conducted, evidence of vaccine efficacy against drug abuse is lacking. Factors contributing to this include low antibody response, un-sustained antibody response, variable antibody response between individuals, and continued drug use despite an antibody response. The underlying reasons are not clear, but could relate to vaccine design, individuals’ genetic makeup, or a combination of both. As captured in the literature, this review also identified the importance of the drug abuser’s personal motivation to discontinue using drugs as part of vaccine-based treatments [1,2,6,11,12,17,19,40,41]. These factors, as well as the complexities associated with addiction, may help to explain the challenges associated with clinical trials for vaccine-based treatments thus far. Therefore, this review also highlights the need for novel treatment strategies in instances where patients do not respond to current treatments, including while the search for efficacious vaccine-based treatments continues. Despite using a comprehensive search strategy, this review found a number of the conducted clinical trials have not been published in peer-reviewed journals. This may be due to insignificant findings. While there has been some suggestion that health professionals—such as pharmacists, nurses, and medical doctors—should be ready to play a role in the delivery of vaccines against drug abuse, at this point in time, without completion of successful phase 3 studies indicating vaccine efficacy, upskilling health professionals in this area is premature.

## Figures and Tables

**Table 1 vaccines-10-00860-t001:** Summary of the details of clinical trials for treatments against nicotine, cocaine, methamphetamine, and opioid abuse [43,44,45,46,47,48,49,50,51,52,53,54,55,56,57,58,59,60,61,62,63,64,65].

Drug	NCT Number	Study Completion Date	Study Type	Phase	Multicenter?	Intervention	Randomized?	Placebo Controlled?	Blinding?	Number of Enrolled Subjects	Defined Secondary Measures?	Status *
Nicotine	NCT01318668	June 2012	Interventional (crossover assignment)	1, 2	N	NicVAX	Y	Y	N	38	Y	c
Nicotine	NCT00218413	August 2006	Interventional (parallel assignment)	2	N	NicVAX	N	N	N	51	Y	c
Nicotine	NCT00995033	September 2012	Interventional (parallel assignment)	2	Y	NicVAX	Y	Y	Double	558	Y	c
Nicotine	NCT00598325	October 2010	Interventional (parallel assignment)	1, 2	N	NicVAX	N	N	N	74	Y	c
Nicotine	NCT00318383	December 2007	Interventional (parallel assignment)	2	Y	NicVAX	Y	Y	Double	313	Y	c
Nicotine	NCT00996034	February 2011	Interventional (single-group assignment)	2	N	NicVAX	N	N	N	14	N	c
Nicotine	NCT01304810	August 2011	Observational (cohort)	3	Y	NicVAX	Y	Y	Double	300	Y	c
Nicotine	NCT01102114	November 2011	Interventional (parallel assignment)	3	Y	NicVAX	Y	Y	Double	1000	Y	c
Nicotine	NCT00836199	July 2011	Interventional (parallel assignment)	3	Y	NicVAX	Y	Y	Double	1000	Y	c
Nicotine	NCT01478893	March 2013	Interventional (parallel assignment)	1	N	SEL-068	Y	Y	Double	82	Y	c
Nicotine	NCT01672645	December 2015	Interventional **	1	N	NIC7-001 and NIC7-003	Y	Y	Double	277	Y	c
Nicotine	NCT00736047	October 2009	Interventional (parallel assignment)	2	Y	NIC002	Y	Y	Double	200	Y	c
Nicotine	NCT01280968	April 2013	Interventional (parallel assignment)	2	N	NIC002	Y	Y	N	52	N	c
Nicotine	NCT00369616	October 2005	Interventional (parallel assignment)	2	N	CYT002-NicQb	Y	Y	N	341	Y	c
Nicotine	NCT00633321	February 2009	Interventional (parallel assignment)	2	Y	TA-NIC	Y	Y	Double	522	Y	c
Cocaine	NCT00965263	August 2009	Interventional (parallel assignment)	2	N	TA-CD	Y	N	N	15	Y	c
Cocaine	NCT00142857	July 2014	Interventional (parallel assignment)	2	Y	TA-CD	Y	Y	Double	115	Y	c
Cocaine	NCT00969878	July 2014	Interventional (parallel assignment)	3	Y	TA-CD	Y	Y	Double	300	Y	c
Cocaine	NCT01846481	June 2013	Interventional (crossover assignment)	2	N	RBP-8000	Y	Y	Double	40	Y	c
Cocaine	NCT01887366	October 2014	Interventional (parallel assignment)	2	Y	TV-1380	Y	Y	Double	208	Y	c
Cocaine	NCT02455479	December 2025 ***	Interventional (sequential assignment)	1	N	dAd5GNE	Y	Y	Double	30	Y	r
Methamphetamine	NCT01603147	July 2013	Interventional (single group assignment)	1	N	Ch-mAb7F9	Y	Y	Double	42	Y	c
Multivalent opioids	NCT04458545	December 2023 ***	Interventional (parallel assignment)	1A/1B	Y	Oxy(Gly)4-sKLH	Y	Y	N	45	N	r

* c = completed, r = recruiting; ** = no assignment specified; *** = estimated study completion date, Y = yes, N = N.

**Table 2 vaccines-10-00860-t002:** Summary of studies for vaccines against nicotine abuse, cocaine, methamphetamines, and opioids [1,6,25,26,27,43,44,45,46,47,48,49,50,51,52,53,54,55,56,57,58,59,60,61,62,63,64,65,66].

Drug of Abuse	Vaccine	Hapten	Carrier	Adjuvant	Trial Phase(s)
Nicotine	NicVAX	3′Amimomethylnicotine	Pseudomonas aeruginosa rEPA	Alum	1, 2, 3
	SEL-068	Nicotine	Proprietary polymer nanoparticle	T-cell-targeting peptide and TLR agonist	1
	NIC7-001 and NIC7-003	5-Aminoethoxy-nicotine and *	Cross-reactive material	Alum	1
	NIC002	O-succinyl-3′-hydroxymethylnicotine	VLP from bacteriophage Qß	Alum	2
	CYT002-NICQb	Nicotine	VLP from bacteriophage Qß	*	2
	TA-NIC	Nicotine N1-butyric acid	rCTB	Alum	2
Cocaine	TA-CD	Succinyl norcocaine	rCTB	Alum	2, 3
	RBP-8000 100 and 200 mg	N/A; metabolizing enzyme therapy			2
	TV-1380 150 and 300 mg	N/A; metabolizing enzyme therapy			2
	dAd5GNE Vaccine	GNE-cocaine	Disrupted adenovirus	Proprietary adjuvant	1
Methamphetamine	Ch-mAB7f9	N/A; chimeric monoclonal antibody			1
Opioids	Oxy(Gly)4-sKLH	Oxycodone-based	Keyhole limpet hemocyanin	Alum	1

* = Unable to locate details; rEPA = recombinant Pseudomonas aeruginosa exotoxin; VLP = virus-like particle; rCTB = recombinant cholera toxin subunit B; TLR = Toll-like receptor; GNE = [6-(2R,3S)-3-(benzoyloxy)-8-methyl-8-azabicyclo [3.2.1] octane-2-carboxoamido-hexanoic acid]; N/A = not applicable.

## Data Availability

Not applicable.

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
