# Peer review of "Vaccines against Drug Abuse—Are We There Yet?"

_vaccines, 2022, doi:10.3390/vaccines10060860_

Round 1

Reviewer 1 Report

The manuscript deals with a relevant area of interest such as the Vaccine use for the treatment of illicit drug abuse. 

They however report negative results coming from CT on the Vaccine development.  It should be of note that currently the medical treatment for opioid-dependent patients is based on treatments with new emerging drugs and psychological supporting care for the patients.   Being a review it is important to cite all recent references related to the novel medical treatments for opioid-dependent patients such as Maqoud et al., Pharmaceutics 2022.

Certainly, the drug treatment is not fully satisfactory and novel strategies are needed for those patients unresponsive to the current drug therapy. This should be emphasized in the discussion sections.

Minor points see page 2 line 48  

Author Response

Response to Reviewer 1 Comments

The manuscript deals with a relevant area of interest such as the Vaccine use for the treatment of illicit drug abuse.

Point 1: They however report negative results coming from CT on the Vaccine development.  It should be of note that currently the medical treatment for opioid-dependent patients is based on treatments with new emerging drugs and psychological supporting care for the patients.   Being a review it is important to cite all recent references related to the novel medical treatments for opioid-dependent patients such as Maqoud et al., Pharmaceutics 2022

Response 1: Please provide your response for Point 1. (in red)

Extra detail to capture this important point has been included in the “Introduction”. As per rows 52 and 53, “In the case of opioid-dependant patients, treatment options also extend to new, emerging drugs and psychological support”. Reference 54, which references the kindly provided Magoud et al., Pharmaceutics 2022 article has also been included.

Point 2: Certainly, the drug treatment is not fully satisfactory and novel strategies are needed for those patients unresponsive to the current drug therapy. This should be emphasized in the discussion sections.

Minor points see page 2 line 48

Response 2: Please provide your response for Point 2. (in red)

Extra detail to capture this important point has been included in the “Conclusion” section of the Abstract, “This review further highlighted the need for novel treatments strategies in instances where patients do not respond to current treatments, and while the search for efficacious vaccine-based treatments continues”. Extra detail is also captured in the “Conclusions” section; “As captured in the literature, this review also identified the importance of the drug abuser’s personal motivation to discontinue using drugs as part of vaccine-based treatments [6, 9, 11, 12, 18, 20, 34, 42, 43]. These factors as well as the complexities associated with addiction, may help to explain the challenges associated with clinical trials for vaccine-based treatments thus far. This review therefore also highlighted the need for novel treatment strategies in instances where patients do not respond to current treatments, including while the search for efficacious vaccine-based treatments continues.”

Reviewer 2 Report

This review revisits the clinical literature concerning the status of the vaccines to treat drug addiction, and includes both legal (i.e. nicotine) and illegal (i.e. cocaine, methamphetamine and opioids) drugs. It is a well written, clear and comprehensive review on an important topic that has important medical implications and a high social cost. Notably, this review fills an important gap in the literature since the last review on vaccines against drug addiction is quite dated.

Below some suggestions to improve clarity and ease of reading:

- In the RESULTS section, please provide a brief definition of phase 1, 2 and 3, this review will be likely read also by students and non-clinicians and clarification of the different steps in clinical trials may result useful.

- Table 1 is quite dense of information: deleting columns with no differences can help the reader to better appreciate the differences among the different trials. In my opinion, that ALL clinical trials indicate inclusion and exclusion criteria can be stated in the text rather than in the table. Similarly, the penultimate column of the table can be reduce to indicate secondary measures only, as ALL clinical trials define primary measures.

- Table 2: all abbreviations should be indicated in the legend, including the most obvious ones (i.e. N/A).

- Adverse effects are indicated for Phase II trial NCT00369616 only: please specify which effects have been reported. Moreover, can authors confirm that all the other cited trials did not report adverse effects?

Author Response

Response to Reviewer 2 Comments

This review revisits the clinical literature concerning the status of the vaccines to treat drug addiction, and includes both legal (i.e. nicotine) and illegal (i.e. cocaine, methamphetamine and opioids) drugs. It is a well written, clear and comprehensive review on an important topic that has important medical implications and a high social cost. Notably, this review fills an important gap in the literature since the last review on vaccines against drug addiction is quite dated.

Below some suggestions to improve clarity and ease of reading:

Point 1: - In the RESULTS section, please provide a brief definition of phase 1, 2 and 3, this review will be likely read also by students and non-clinicians and clarification of the different steps in clinical trials may result useful.

Response 1: Please provide your response for Point 1. (in red)

Rows 102 to 113 in the “Results” section capture this very useful suggestion, “Phase 1, 2 and 3 clinical trials were included in this review. “Phase 1” refers to “a phase of research to describe clinical trials that focus on the safety of a drug, usually conducted with healthy volunteers, and usually involving a small number of participants” [44]. “Phase 2” refers to “a phase of research to describe clinical trials that gather preliminary data on whether a drug works in people with a certain disease/condition. Safety continues to be a focus [44]. “Phase 3” refers to “a phase of research to describe clinical trials that gather more information about a drug’s safety and effectiveness by studying different populations and different dosages and by using the drug in combination with other drugs. These studies typically involve more participants” [44].

Point 2: - Table 1 is quite dense of information: deleting columns with no differences can help the reader to better appreciate the differences among the different trials. In my opinion, that ALL clinical trials indicate inclusion and exclusion criteria can be stated in the text rather than in the table. Similarly, the penultimate column of the table can be reduce to indicate secondary measures only, as ALL clinical trials define primary measures.

Response 2: Please provide your response for Point 2. (in red)

Rows 102 to 104 in the “Results section” capture this very useful suggestion, “Defined inclusion criteria, exclusion criteria and primary outcome measures were included for each clinical trial, with defined secondary outcome measures included for all but 3 of the trials [see Table 1]”. In table 1, “Defined inclusion & exclusion criteria” column?” was removed and replaced with “Defined secondary measures?”. For completeness, a “Study Completion Date” column and associated details for each trial has been added to table 1.

Point 3: - Table 2: all abbreviations should be indicated in the legend, including the most obvious ones (i.e. N/A).

Response 3: Please provide your response for Point 2. (in red)

“N/A” entry in table 2 replaced with “T cell -targeting peptide and TLR agonist”. Definitions for “TLR”, “GNE” and “N/A” have also been included in legend. “*** = estimated Study Completion Date” was also added to the legend.

Point 4: - Adverse effects are indicated for Phase II trial NCT00369616 only: please specify which effects have been reported. Moreover, can authors confirm that all the other cited trials did not report adverse effects?

Response 4: Please provide your response for Point 2. (in red)

Studies for which findings were able to be located and which included “adverse event” reporting have been included in the manuscript. Please refer to the “Discussion” section for the following clinical trials: NCT01318668, NCT00995033, NCT00996034, NCT00836199, NCT01102114, NCT01280968, NCT00369616 (additional detail), NCT00142857 and NCT00969878.

Reviewer 3 Report

This is an interesting and clearly written study summarizing the effects of vaccines against drugs of abuse. The analyzed data are from registered and mostly completed clinical trials and the results are clear. At least clear in terms of the trials that were completed which are mostly against nicotine and here they show no longterm effect.

Although slclear and well done, parts of this review leave the reader wondering;

  1. If vaccines have been looked at for so long yet are so inconclusive in terms of longterm benefits, can the authors add some discussion about this. Are we simply not targeting the right molecules or using the right approach or is the modality lacking? Is there any instance or approach that vaccines could be useful?
  2. The authors state that this study assesses the effects of vaccines for drugs of abuse yet the conclusive data are about nicotine. Should the title and emphasis of the review be altered to reflect this?
  3. This review looks at the effects of targeting the drug per se and not any other molecules. This should be made clear and the studies that are nor the subject of the review removed or the review expanded to include these studies.

Author Response

Response to Reviewer 3 Comments

This is an interesting and clearly written study summarizing the effects of vaccines against drugs of abuse. The analyzed data are from registered and mostly completed clinical trials and the results are clear. At least clear in terms of the trials that were completed which are mostly against nicotine and here they show no longterm effect.

Although clear and well done, parts of this review leave the reader wondering;

  1. Point 1: - If vaccines have been looked at for so long yet are so inconclusive in terms of longterm benefits, can the authors add some discussion about this. Are we simply not targeting the right molecules or using the right approach or is the modality lacking? Is there any instance or approach that vaccines could be useful?

Response 1: Please provide your response for Point 1. (in red)

This very important aspect has now been captured in rows 6 to 13 in the “Conclusions”, “As captured in the literature, this review also identified the importance of the drug abuser’s personal motivation to discontinue using drugs as part of vaccine-based treatments [6, 9, 11, 12, 18, 20, 34, 42, 43]. These factors as well as the complexities associated with addiction, may help to explain the challenges associated with clinical trials for vaccine-based treatments thus far. This review therefore also highlighted the need for novel treatment strategies in instances where patients do not respond to current treatments, including while the search for efficacious vaccine-based treatments continues.

Point 2: - The authors state that this study assesses the effects of vaccines for drugs of abuse yet the conclusive data are about nicotine. Should the title and emphasis of the review be altered to reflect this

Response 2: Please provide your response for Point 2. (in red)

The scope of this review was to research legal and illicit drugs of abuse. Undertaking this review revealed differing levels of research and clinical trial progress for different drugs which was in itself insightful, considered relevant and of interest to the reader, hence the inclusion and mention of all clinical trials including whether they were recruiting, completed etc.

Point 3: This review looks at the effects of targeting the drug per se and not any other molecules. This should be made clear and the studies that are nor the subject of the review removed or the review expanded to include these studies.

Response 3: Please provide your response for Point 2. (in red)

This review focused on vaccine-based options for treating a range of drugs of abuse. The review revealed 2 of the cocaine trials were enzyme-based (and so out of scope), and the methamphetamine trial was monoclonal antibody based (also out of scope). For completeness, these discoveries were included in the “Discussion” section. Additional qualifying comments were also captured in the “Discussion” section; “Phase II trials NCT01846481 and NCT01887366 were completed for RBP-8000 and TV-1380 [44]. These trials were enzyme based and were not the focus of this review”.

Phase I trial NCT01603147 for ch-mAb7f9 was completed [52]. This trial was monoclonal antibody based and was not the focus of this review”. Details were also included in table 2 to capture this: including “N/A; metabolising enzyme therapy” and “N/A; chimeric monoclonal antibody” as applicable.
